# Optimization of Platelet-Rich Plasma Preparation for Regenerative Medicine: Comparison of Different Anticoagulants and Resuspension Media

**DOI:** 10.3390/bioengineering11030209

**Published:** 2024-02-23

**Authors:** Alexandra Carvalho, Ana Filipa Ferreira, Maria Soares, Susana Santos, Patrícia Tomé, Juliana Machado-Simões, Ana Sofia Pais, Ana Paula Sousa, Artur Paiva, Teresa Almeida-Santos

**Affiliations:** 1Reproductive Medicine Unit, Gynecology, Obstetrics, Reproduction and Neonatology Department, Centro Hospitalar e Universitário de Coimbra, 3004-561 Coimbra, Portugal; 10494@chuc.min-saude.pt (A.F.F.); 10728@chuc.min-saude.pt (A.S.P.); anasousa@chuc.min-saude.pt (A.P.S.); teresaalmeidasantos@chuc.min-saude.pt (T.A.-S.); 2CICS-UBI-Health Sciences Research Centre, University of Beira Interior, 6200-506 Covilhã, Portugal; 3CNC-UC—Center for Neuroscience and Cell Biology, University of Coimbra, 3004-504 Coimbra, Portugal; mms2596@gmail.com (M.S.); julianasmsimoes@gmail.com (J.M.-S.); 4Faculty of Medicine, University of Coimbra, 3000-548 Coimbra, Portugal; 5Doctoral Programme in Experimental Biology and Biomedicine (PDBEB), Institute for Interdisciplinary Research, University of Coimbra, 3000-456 Coimbra, Portugal; 6Flow Cytometry Unit, Clinical Pathology Department, Centro Hospitalar e Universitário de Coimbra EPE, 3000-075 Coimbra, Portugal; susana.almeida.santos@chuc.min-saude.pt (S.S.); patricia.tome88@gmail.com (P.T.); artur.paiva@chuc.min-saude.pt (A.P.); 7Department of Life Sciences, University of Coimbra, 3000-456 Coimbra, Portugal; 8Center for Innovative Biomedicine and Biotechnology (CIBB), Institute for Clinical and Biomedical Research (iCBR), University of Coimbra, 3000-548 Coimbra, Portugal; 9Ciências Biomédicas Laboratoriais, ESTESC-Coimbra Health School, Instituto Politécnico de Coimbra, 3046-854 Coimbra, Portugal

**Keywords:** platelet-rich plasma, regenerative medicine, anticoagulants, growth factors

## Abstract

Platelet-rich plasma (PRP) has emerged as a promising therapy in regenerative medicine. However, the lack of standardization in PRP preparation protocols presents a challenge in achieving reproducible and accurate results. This study aimed to optimize the PRP preparation protocol by investigating the impact of two different anticoagulants, sodium citrate (SC) and ethylenediaminetetraacetic acid (EDTA), and resuspension media, plasma versus sodium chloride (NaCl). Platelet recovery rates were calculated and compared between groups, in addition to platelet activity and vascular endothelial growth factor (VEGF) released into plasma after PRP activation. The platelet recovery rate was higher with EDTA in comparison to SC (51.04% vs. 29.85%, *p* = 0.005). Platelet activity was also higher, with a higher expression of two platelet antibodies, platelet surface P-Selectin (CD62p) and PAC-1, in the EDTA group. The concentration of VEGF was higher with SC in comparison to EDTA (628.73 vs. 265.44 pg/mL, *p* = 0.013). Platelet recovery rates and VEGF levels were higher in PRP resuspended in plasma when compared to NaCl (61.60% vs. 48.61%, *p* = 0.011 and 363.32 vs. 159.83 pg/mL, *p* = 0.005, respectively). Our study reinforces the superiority of EDTA (as anticoagulant) and plasma (for resuspension) in obtaining a higher platelet recovery and preserving platelet functionality during PRP preparation.

## 1. Introduction

The therapeutic potential of platelet-rich plasma (PRP) was first reported in the 1970s to treat thrombocytopenia [1]. Since the 1990s, it has been extensively applied in regenerative medicine, particularly in the recovery from athletic injuries [2,3]. The theory underlying this treatment is the natural capacity of platelets to induce cell proliferation, revascularization, and tissue repair due to the various growth factors contained in the platelet α granules [2].

Due to its effectiveness on tissue regeneration and repair, PRP has been progressively applied in other medical fields, such as orthopedics, dentistry, neurosurgery, ophthalmology, urology, dermatology, cardiology, gynecology, and veterinary medicine [4]. PRP is an autologous therapy prepared from the whole blood of a patient, providing a blood-derived product with a platelet concentration at least two times higher than concentration of the whole blood [5,6,7].

The great potential of this product derives from the platelets’ α-granules content in cytokines and growth factors, namely platelet-derived growth factor (PDGF), insulin-like growth factor (IGF), vascular endothelial growth factor (VEGF), platelet-derived angiogenic factor (PDAF), transforming growth factor beta (TGF-β), fibroblast growth factor (FGF), epidermal growth factor (EGF), connective tissue growth factor (CTGF), and interleukin-8 (IL-8) [8,9,10]. These factors have demonstrated a role in tissue repair and were also able to stimulate the migration and differentiation of mesenchymal stem cells (MSC). Once platelets are activated by endogenous factors (e.g., thrombin, collagen, fibronectin) or exogenously (e.g., contact with glass), α-granules release multiple growth factors [11]. Those factors start to be released within 10 min and it is expected that 70–95% would be released around 1 h after platelet activation [12,13]. To ensure that growth factors are only set free in the target tissue, platelet aggregation and activation should not occur during blood collection and PRP preparation, even though it remains unclear if PRP activation should be performed prior to its application and which activator to employ [14].

The protocols of PRP preparation were initially described in 1999 [15], and since then, a vast number of different protocols have been described, considering the multiple options of centrifugation forces and time, centrifugal acceleration, temperature, anticoagulants, and volume of processed blood. These protocols aim to optimize platelet isolation and concentration. According to the therapeutic purpose, it is possible to selectively isolate specific constituents. By employing varied preparation techniques, it becomes feasible to procure samples enriched in platelets alone, platelets alongside leukocytes, platelets coupled with fibrin, or a combination of leukocytes, platelets, and fibrin [16]. It is crucial to note that all these factors, including the strength and duration of centrifugation, the speed of centrifugal force applied, temperature conditions, the type of anticoagulants used, and the amount of blood processed, have the potential to significantly impact both the concentration and activation state of platelets [17].

Despite the diverse number of protocols utilized for the preparation of PRP, all involve blood collection, initial centrifugation to separate red blood cells, subsequent centrifugations to concentrate platelets, and, optionally, platelet activation. Usually, around 3 to 5 cubic centimeters (cc) of PRP can be extracted from a 30cc blood draw, although this quantity can vary based on the individual’s platelet count, the specific device used, and the method applied. The blood collection procedure entails incorporating an anticoagulant to inhibit platelet activation. PRP could be prepared reliably and cost-effectively by simply isolating the blood constituents according to their density using a commercial kit or following a two-step centrifugation protocol [16,17]. The first centrifugation step is conducted at a constant acceleration to separate red blood cells from the remaining whole blood volume. After the first centrifugation, it is possible to distinguish three layers: a top layer of plasma (platelet-poor plasma), a middle thin layer of leukocytes and platelets (buffy coat) and a lower layer of erythrocytes. The top layer of plasma (with or without the buffy coat) is collected and proceeded for a second centrifugation to increase the platelet concentration. The second centrifugation step aims to concentrate platelets. Subsequently, the upper portion of the volume, predominantly composed of platelet-poor plasma is removed. The pellets are then homogenized in the smallest final plasma volume. Typically, the PRP is diluted in the lower one-third to create the PRP [16,17]. The pellet is resuspended in a media, that could be plasma [1,18,19] or sodium chloride (NaCl) [4]. Factors influencing PRP yield include blood draw, centrifugation parameters, and choice of anticoagulants [16].

Selecting an appropriate anticoagulant is crucial to maintain the optimal functionality, integrity, and morphology of platelets. Regarding the choice of anticoagulant, most experts advise against using Ethylenediaminetetraacetic acid (EDTA) due to its potential to harm the platelet membrane. Instead, anticoagulants containing citrate and dextrose or sodium citrate (SD) are recommended for preserving platelet quality. EDTA, heparin, acid citrate dextrose (ACD) and SC have been described for blood collection prior to PRP preparation due to their capacity to avoid clot formation and their documented use for hematological studies, hemostasis, and coagulation tests [20]. The majority of commercial kits use SC and ACD [16] to prepare PRP, but there is no agreement on the most effective anticoagulant for this purpose since studies have shown contradictory results regarding the platelet recovery rate and growth factors [8,16,20,21,22,23,24].

Flow cytometry is a recognized method for measuring platelet concentration (platelet count) and characterizing platelet activation (CD62p, anti-CD42b, PAC1) directly in the whole blood. This cytometric analysis of platelets allows the assessment of platelet function by the utilization of different fluorochrome–antibody conjugates. This recent methodological approach demonstrates several advantages, such as the minimal volume of total blood requirement, the analysis of platelets in their physiological environment, and minimal sample manipulation. Additionally, it also allows the determination of the activation and reactivity state of circulating platelets to agonists [25]. Testing can also be carried out in platelet-rich plasma or washed platelets [26]. Specific components derived from platelets are employed for flow cytometry testing purposes. CD62P (platelet surface P-Selectin) is mobilized from the α-granules to the surface when platelet activation occurs to promote platelet–platelet and platelet–fibrin binding [26]. CD42b (Glycoprotein Ib) is a component of the platelet GPIb-V-IX complex, which binds to von Willebrand factor, facilitating platelet adhesion after their activation [26]. PAC1 is directed against activated GP-IIb/IIIa on activated platelets [27]. Therefore, the flow cytometric detection of these antigens is a reliable method to explore platelet activation [28].

As mentioned, the preparation of the PRP could follow different methods, and there is no consensus on the best protocol to obtain PRP for clinical application. The steps to obtain PRP have to be accurately chosen as they directly affect the platelet concentration and viability, that consequently influence the concentration of growth factors in the target tissue and the efficacy of the PRP therapy [16,17].

Therefore, our objective was to explore the utilization of diverse anticoagulants and resuspension media in the preparation of PRP by conducting a comparative analysis of the platelet recovery rate, the percentage of active platelets, and the concentration of platelet-derived VEGF. Ultimately, this study aimed to provide evidence for the selection of these parameters during the establishment of a protocol for PRP preparation.

## 2. Materials and Methods

This study was designed, conducted, and reported in accordance with the principles of Good Clinical Practice guidance and with the 1964 Helsinki declaration and its later amendments. Ethical approval was granted by the Hospital Ethics Board (reference number CHUC-171-20), and written consent was obtained from all participants.

The research was carried out at the Reproductive Medicine Unit of a tertiary hospital. Women planning to undergo frozen embryo transfer (FET) were informed about the study. As the FET requires measuring plasma progesterone levels on the day of transfer, the patients who gave the written informed consent also donated blood for the study, in the same procedure.

This study employs a two-step approach to investigate the impact of different anticoagulants and PRP resuspension media on PRP preparation. In the first experiment, we scrutinized the efficacy of SC (Sarstedt S-monovette^®^, Nümbrecht, Germany) versus EDTA (Sarstedt S-monovette^®^, Nümbrecht, Germany) as anticoagulants for blood collection. Blood samples were drawn from 21 participants, with each individual providing a 28 mL venous blood sample. Half of this volume was collected in tubes containing SC, while the other half was collected in tubes containing EDTA. This dual-collection method ensured consistency by using samples from the same patients.

According to the results obtained in the first experiment, we proceeded to the second experiment. Here, we compared plasma and NaCl as PRP resuspension media. Similarly to the first experiment, blood samples were obtained from 19 participants using the same collection technique. Samples collected in EDTA-containing tubes were utilized for this phase. Half of the volume from each participant was used to prepare PRP resuspended in plasma, while the other half was utilized for PRP resuspended in NaCl.

Throughout the study, meticulous attention was paid to maintaining controlled conditions. Peripheral blood samples were collected by two trained healthcare professionals using a 21G butterfly catheter affixed to negative pressure-receiving tubes. This standardized approach minimized potential variables during sample collection, ensuring the reliability and validity of the study findings. Additionally, this approach has been chosen to help prevent clotting and unintended platelet activation.

A small amount of the blood sample (1 mL) was separated to perform the initial evaluation of whole blood by flow cytometry and the remaining volume was transferred to two empty 15 mL centrifugal tubes (Sarstedt S-monovette^®^, Nümbrecht, Germany) and centrifuged at 300× *g* for 10 min. The upper layer (plasma and buffy coat) was then transferred to a new empty sterile tube and centrifuged at 700× *g* for 15 min, to pellet the platelets. The supernatant was discarded, and the pellet was resuspended in 1:4 volume with the lower portion of plasma in the first experiment. In the second experiment, we sought to compare two different resuspension solutions and the pellet was resuspended in (1) plasma or (2) NaCl (0.9%, B.Braun, Queluz de Baixo, Portugal). The entire procedure was performed at room temperature (21–24 °C).

A schematic representation of the PRP preparation is presented in Figure 1.

After preparing a whole blood sample of 1 mL and PRP samples, we promptly transported them to the hospital’s cytometry unit, avoiding agitation and maintaining a temperature range of 21–24 °C.

To assess platelet recovery rate, the total platelet cell count was determined in whole blood and PRP samples. Platelet count was performed in a ABX Micro ES60 (Horiba, Japan) hemacytometer.

The platelet recovery rate (PRR) was determined by calculating the ratio of the platelet concentration in the PRP sample to the platelet concentration in the whole blood sample, considering the respective volumes This calculation was performed using the formula:(1)$$PRR(\%)=\frac{Platelets\;concentration\;in\;PRP\times volume\;of\;PRP}{Platelets\;concentration\;in\;whole\;blood\times volume\;blood}\times 100$$

The adoption of PRR as a reporting metric enhances the clarity and interpretability of our findings within the context of PRP research. We chose this method to best capture the relative increase in platelet concentration within PRP compared to the baseline concentration in whole blood. Additionally, PRR accommodates the inherent variability present in biological samples across individuals, as it anchors the analysis to the initial platelet count.

To assess the platelet activity state of PRP preparation, we evaluated the presence of specific markers of platelet activation (CD62p and PAC1) before and after inducing platelet activation with thrombin.

For the platelet activity assay, samples were prepared by diluting the PRP samples in NaCl (0.9%) at a ratio of 1:10. PRP samples were mixed with 5 µL of each monoclonal antibody, CD62pPE and PAC1-FITC and analyzed using a flow cytometer. Identical PRP samples were incubated with 6.5 µL of thrombin (50 µM) (Sigma, Merck, Darmstadt, Germany). The incubation was carried out at room temperature, in a dark environment, for a duration of 15 min. Following this incubation period, samples were resuspended in 100 µL of NaCl (0.9%) and analyzed using a FACSCalibur flow cytometer (BD Biosciences, USA). Results were expressed as the percentage of CD62p positive or PAC-1 positive platelets after in vitro activation.

After activation with thrombin, PRP samples were centrifuged at 1200× *g* for 10 min and the supernatant was stored at −80 °C. The concentration of VEGF released into the plasma after PRP activation was measured using an ELISA (Enzyme-Linked Immunosorbent Assay). The ELISA kit used was the Human VEGF Immunoassay by R&D Systems^®^ (RD systems Quantikine ELISA, Minneapolis, MN, USA). The sensitivity of the kit was determined by the minimum detectable dose of human VEGF, which is <9.0 pg/mL. The ELISA procedure followed the manufacturer’s instructions without any modifications. Briefly, the concentrations of VEGF in the samples were determined by comparing the optical densities of the samples to the standard curve generated using the known concentrations of VEGF standards provided in the kit. To ensure the accuracy and reliability of the ELISA results, several controls were used. The VEGF quantification assays were performed in duplicate for all the PRP preparation samples.

Values are expressed as median and interquartile range (Q1–Q3). Variables were analyzed using paired Wilcoxon test, since samples for comparison of the anticoagulants (SC versus EDTA) and resuspension solutions (plasma versus NaCl) were obtained from the same patient. Statistical analysis was performed with the support of IBM SPSS Statistics for Windows, (Version 28.0. IBM Corp, Armonk, NY, USA), with the level of significance fixed at 5%.

## 3. Results

To compare different anticoagulants, blood samples were collected from 26 patients. One patient was excluded due to a previously unknown medical condition affecting platelets. Five patients were excluded due to insufficient blood volume from one of the anticoagulants. Therefore, anticoagulants were compared by analyzing blood samples from 21 patients. To compare different resuspension media, blood samples were collected from 19 patients.

### 3.1. Platelet Concentration

The mean platelet concentration in whole blood was 200,190 cells/mm^3^ (range, 117,000 to 347,000) for samples collected in tubes containing SC, and 200,190 cells/mm^3^ (range, 86,000 to 406,000) for those collected in tubes containing EDTA. The mean platelet concentration in PRP was 1,858,410 cells/mm^3^ (range, 102,000 to 4,260,000) for SC-collected samples, and 3,458,222 cells/mm^3^ (range, 1,476,000 to 5,322,000) for EDTA-collected samples. The platelet concentration increased 9 to 17 times when transitioning from whole blood to PRP.

### 3.2. Platelet Recovery Rate

PRP obtained from blood samples collected from the same patient with different anticoagulants had significantly different platelet recovery rates (*p* = 0.005), being higher when EDTA was used to collect the blood. The median recovery rate was 51.04% with EDTA versus 29.85% with SC (Table 1, Figure 2A).

Platelet recovery rates were higher (*p* = 0.011) in PRP resuspended in plasma (61.60%) when compared to NaCl (48.61%) (Table 2, Figure 2B).

### 3.3. Platelet Activity Assay of CD62p and PAC-1

The expression of platelet activity markers was significantly higher in PRP obtained when EDTA was used to collect the blood. Expression of CD62p and PAC-1 were 39.60 versus 16.20 (*p* = 0.035) and 79.20 and 55.50 (*p* = 0.014) in the EDTA versus SC groups, respectively (Table 1).

There was no difference in the expression of CD62p and PAC-1 when different resuspended solutions were compared (Table 2).

### 3.4. Determination of VEGF Concentration

The concentration of VEGF released into plasma after PRP activation was significantly higher in samples collected with SC, with a median of 628.73 pg/mL, when compared to EDTA, with a median of 265.44 pg/mL (*p* = 0.013) (Table 1).

Resuspension of PRP in plasma had higher levels of VEGF, with a median of 363.32 pg/mL, when compared to NaCl, with a median of 159.83 pg/mL (*p* = 0.005) (Table 2).

## 4. Discussion

Our findings suggest that the use of EDTA for blood collection is better than SC when considering the platelet recovery rate and platelet viability (i.e., the potential to be activated when desired) of the PRP. Despite the significantly lower VEGF concentration with EDTA in comparison to citrate, both PRP preparations had high VEGF levels. In addition, resuspension of PRP in plasma seems to be preferred due to a higher platelet recovery rate and a higher VEGF concentration.

When preparing PRP from whole blood, anticoagulants are used to prevent blood clotting and preserve the platelets in a nonactivated state, therefore preventing the release of growth factors before exposure to target tissue. The choice of the anticoagulant can influence the platelet recovery rates in PRP preparation. EDTA and citrate are commonly used anticoagulants in PRP preparation both acting by chelation of calcium. Acid citrate dextrose (ACD) is often used to preserve blood for transfusion, since dextrose support blood cell metabolism and viability [29].

We had a lower platelet recovery rate in relation to other published studies [8,21]. Despite, our protocol of PRP preparation yielded a 9- to 17-fold concentration of platelets, which is high and in accordance with the platelet concentrations reported by other authors [8,21,30]. Importantly, we had a higher platelet recovery rate and a higher expression of platelet activity markers when EDTA was used, a finding also reported by others [21,24]. This can be attributed to the ability of EDTA to maintain platelets in a more natural state without altering their characteristics during the preparation process. Curiously, Amaral et al. found a significantly lower platelet recovery rate with ACD, the anticoagulant of several commercial kits for PRP preparation [20]. However, it is important to note that the optimal anticoagulant choice may vary depending on the specific goals of the PRP application and the intended use, with some authors reporting better results with citrate [31].

The level of VEGF in PRP preparations demonstrated a large variability between individuals and study protocols, ranging from 127 to 1300 pg/mL [23,30,32,33]. It would be expected that a PRP with a higher platelet recovery would also result in a PRP with higher concentration of growth factors after platelet activation. Surprisingly, in our study, the median concentration of VEGF released into plasma with citrate was higher than with EDTA (628.73 pg/mL versus 265.44 pg/mL). This suggests that EDTA may adversely affect platelet degranulation. In fact, the presence of anticoagulant when PRP is being prepared may affect the release and/or measurement of growth factors by platelets. A higher concentration of several growth factors has been reported when using a half-dose of SC [23,34]. Ulasli and colleagues [23] compared the VEGF levels of fresh PRP prepared with a standard dose of SC (140.7 pg/mL), half-dose of SC (240.2 pg/mL) and without the use of any anticoagulant (1303 pg/mL), concluding that the use and dose of SC affects the growth factor composition. In addition, like in our study, they found no correlation between platelet concentration and growth factor concentration, since the protocol with a standard dose of SC had the highest platelet recovery and the lowest VEGF concentration.

On the other hand, a higher concentration might not result in a better therapeutic effect since in vitro studies have shown that the dose–response curves of most growth factors are not linear [29]. Thus, the optimal dose of VEGF and other growth factors may vary depending on the target tissue and intended therapeutic effect.

When considering the resuspension of platelets in PRP, both NaCl and plasma can be used as resuspension media, as they both prevent unintentional platelet activation [29]. Ultimately, the choice between NaCl and plasma as resuspension media depends on the desired outcomes, specific clinical application, and available resources. While resuspending in NaCl is simpler and more standardized, it lacks the biological components present in plasma, such as cytokines and growth factors, that may contribute to the desired therapeutic effects.

In this regard, as expected, we found a significantly higher level of VEGF when the platelet pellet was resuspended in plasma (363.32 pg/mL) in comparison to NaCl (159.83 pg/mL). In addition, we also found a higher platelet recovery rate when plasma was used as the resuspension media, indicating that plasma might provide a more suitable environment for platelet stability and functionality, as suggested by Etulain et al. [35].

Numerous methods are outlined in the literature to prepare PRP, emphasizing the importance of each laboratory devising a standardized procedure. A homemade PRP can be reliably and affordably prepared without relying on commercial kits, but it is crucial to adjust the parameters involved in PRP preparation, as they can influence the concentration and viability of platelets. Among the determinant parameters involved in PRP preparation, the time and centrifugation forces vary according to the therapeutic target, with customization required to optimize platelet concentration for specific applications. However, the choice of anticoagulant used and the selection of resuspension media stand as pivotal initial and final steps, respectively, irrespective of the intended use of PRP.

Many different factors, such as centrifuge type, temperature, duration, harvesting technique, use of activators, etc., may affect the composition of PRP preparations, which may hamper the interpretation of PRP therapies. In addition, there is a large variability between individuals regarding blood cell counts and growth factors concentration. Therefore, we collected blood from a relatively high number of patients (*n* = 21) and compared samples from the same patient maintaining the same procedure except for the anticoagulant used (in the first experiment) and the resuspension media (in the second experiment), supporting the interpretation of our results.

It is worth noting that VEGF is just one of the numerous growth factors and cytokines present in PRP. Therefore, the conclusions based on VEGF quantification represent a limitation of our study. Also, we have not determined the white blood cells count on PRP, which might affect VEGF levels, since it is released by both platelets and leucocytes.

Further research is required to evaluate the impact of anticoagulants and resuspension media on the release and concentration of other growth factors and their overall effect on the therapeutic potential of PRP in regenerative medicine.

## 5. Conclusions

Our study underscores the effectiveness of EDTA in achieving superior platelet recovery rates and maintaining platelet functionality during PRP preparation, while also advocating for the utilization of plasma as a resuspension medium. Despite demonstrating clear advantages over SC in terms of platelet recovery and viability yielding, EDTA-prepared samples revealed lower VEGF concentrations. Additionally, resuspending PRP in plasma is favored due to its association with higher platelet recovery rates and VEGF concentrations.

These findings underscore the critical significance of choosing both the anticoagulant and resuspension medium in PRP preparation, providing guidance for clinicians in selecting appropriate protocols. Whether a higher platelet recovery and potential to be activated is associated with a higher concentration of growth factors and therapeutic benefit still needs further investigation and might depend on the target tissue.

Furthermore, we acknowledge the importance of exploring the stability of PRP prepared using various anticoagulants over extended durations, particularly regarding their clinical applicability. Future investigations should prioritize assessing the long-term stability of PRP preparations, taking into account factors such as platelet integrity, growth factor retention, and overall therapeutic efficacy. These endeavors are crucial for refining PRP preparation protocols and advancing their clinical implementation, ultimately enhancing patient outcomes in regenerative medicine applications.

## Figures and Tables

**Figure 1 bioengineering-11-00209-f001:**
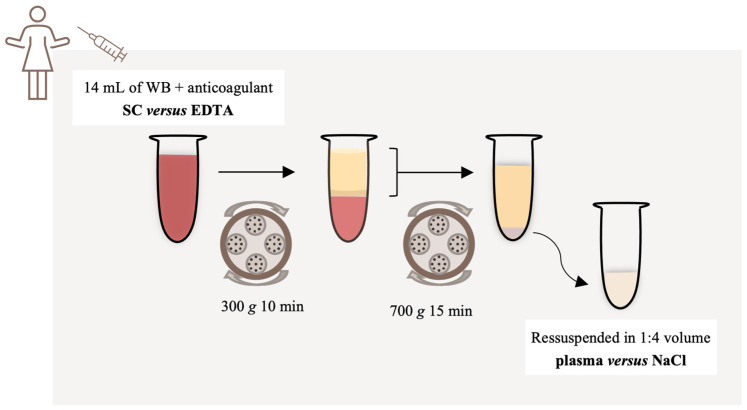
Schematic representation of the protocol to obtain platelet-rich plasma (PRP). Whole blood was collected into tubes with anticoagulant, sodium citrate (SC) versus ethylenediaminetetraacetic acid (EDTA). The samples (14 mL) were transferred to empty tubes and centrifuged at 300× *g* for 10 min. The upper layer (plasma and buffy coat) was then transferred to a new empty sterile tube and centrifuged at 700× *g* for 15 min to pellet the platelets. The supernatant was discarded, and the pellet was resuspended in 1:4 volume of plasma versus sodium chloride (NaCl). SC: sodium citrate; NaCl: sodium chloride; WB: whole blood.

**Figure 2 bioengineering-11-00209-f002:**
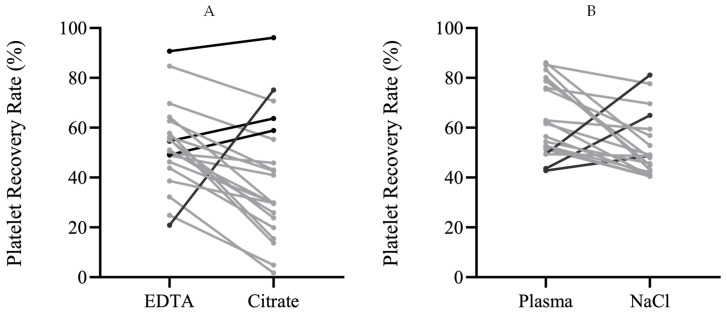
(**A**) Platelet recovery rate using different anticoagulants. Each line represents a patient, connecting the values of platelet recovery rate (PRP) obtained with EDTA versus sodium citrate. Positive ranks (PRP with citrate higher than with EDTA) are depicted in black and negative ranks (PRP with citrate lower than with EDTA) are depicted in grey; (**B**) platelet recovery rate using different resuspension solutions. Each line represents a patient, connecting the values of platelet recovery rate (PRP) obtained with plasma versus NaCl. Positive ranks (PRP with NaCl higher than with plasma) are depicted in black and negative ranks (PRP with NaCl lower than with plasma) are depicted in grey.

**Table 1 bioengineering-11-00209-t001:** Comparison of platelet-rich plasma (PRP) obtained after blood sample collection with different anticoagulants: sodium citrate versus ethylenediaminetetraacetic acid (EDTA).

	Sodium Citrate (*n* = 21)	EDTA (*n* = 21)	*p*-Value
Platelet recovery rate (%)	29.85 (21.82–57.10)	51.04 (45.04–60.24)	0.005
CD62p (%) platelets	16.20 (9.25–36.95)	39.60 (32.10–51.88)	0.035
PAC-1 (%) platelets	55.50 (31.41–68.45)	79.20 (54.85–88.20)	0.014
VEGF (pg/mL)	628.73 (291.52–1100.43)	265.44 (159.84–673.43)	0.013

Values are expressed as median (Q1–Q3). VEGF: vascular endothelial growth factor.

**Table 2 bioengineering-11-00209-t002:** Comparison of PRP resuspension solution with plasma versus NaCl.

	Plasma (*n* = 19)	NaCl (*n* = 19)	*p*-Value
Platelet recovery rate (%)	61.60 (51.29–78.73)	48.61 (42.12–59.43)	0.011
CD62p (%) platelets	24.60 (14.95–42.12)	36.30 (20.80–45.23)	0.314
PAC-1 (%) platelets	65.85 (40.06–85.67)	76.22 (49.71–92.70)	0.159
VEGF (pg/mL)	363.32 (129.56–926.59)	159.83 (116.18–435.82)	0.005

Values are expressed as median (Q1–Q3). VEGF: vascular endothelial growth factor.

## Data Availability

The datasets used and/or analyzed during the current study are available from the corresponding author on reasonable request.

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
