# Peer review of "Optimization of Platelet-Rich Plasma Preparation for Regenerative Medicine: Comparison of Different Anticoagulants and Resuspension Media"

_bioengineering, 2024, doi:10.3390/bioengineering11030209_

Round 1
Reviewer 1 Report
Comments and Suggestions for Authors
The paper is well written.
My concerns refer to the topic that is not novel
Author Response
Dear Reviewer,
Thank you for providing valuable feedback. We appreciate your acknowledgment of the clarity of our paper. Addressing your concern about the novelty of the topic, we would like to offer further clarification.
While it's true that platelet-rich plasma (PRP) has been extensively studied, the field has evolved significantly, resulting in a diverse number of protocols for PRP preparation.
These protocols vary in factors such as the anticoagulant used for blood collection, amount of blood processed, centrifugation forces and duration, temperature, etc. All of these factors can impact PRP composition and ultimately the clinical outcomes, when applied for regenerative medicine. Implementing a standardized PRP preparation protocol is essential to ensure consistency and accuracy in results across studies Therefore, while the concept of PRP itself may not be novel, the focus of our study lies in addressing the crucial need for standardization in PRP preparation protocols.
Reviewer 2 Report
Comments and Suggestions for Authors
In this protocol paper, the authors present the method for preparation of platelets rich plasma and also show the comparison of different anticoagulants. Using simple protocols, the authors present a nice report on preparation of platelets rich anti-coagulant loaded plasma that could have multiple applications in regenerative medicines and organ transplant. I have 2 small suggestions:
1. the authors might also want to compare the stability of different anticoagulants loaded plasma for few days time.
2. figures 2 and 3 could be merged into a single figure.
Author Response
Dear Reviewer,
Thank you for your insightful suggestion. We appreciate your interest in further exploring the stability of different anticoagulants in platelet-rich plasma (PRP) over time.
In our study, we focused on analysing platelets without the influence of platelet pre-activation and assessing the availability of growth factors (GFs). While stability over time is indeed a critical aspect to consider in the clinical application of PRP therapy, our primary objective was to evaluate the effects of different anticoagulants on PRP quality and composition.
However, we acknowledge the importance of investigating the stability of PRP prepared with various anticoagulants over time, especially regarding its clinical implementation. Future research may benefit from assessing the long-term stability of PRP preparations, considering factors such as platelet integrity, growth factor retention, and overall efficacy.
We appreciate your suggestion and will consider it for future investigations in this field. Furthermore, we have incorporated your insights into the conclusion section of the paper.
In response to your recommendation, we have merged Figures 2 and 3 as per your suggestion.
Once again, we extend our gratitude for your contribution to the refinement of our work.
Reviewer 3 Report
Comments and Suggestions for Authors
In this work, the authors claimed that the combination of EDTA as anticoagulant and plasma for resuspension could obtain a higher platelet recovery and preserving platelet functionality during PRP preparation. The following issues should be addressed.
1) the shortcomings of this pair should also be discussed.
2) the presentation of the data should be improved, such as Figure 2, Figure 3.
3) the experimental data is not enough to support the conclusion, for example, the platelet yield , etc., should also be provided.
Author Response
Dear Reviewer,
Thank you for your valuable suggestions regarding our scientific paper.
1) We apologize if there was any lack of clarity in the paper regarding our methodology. Our study aimed to elucidate the advantages of utilizing ethylenediaminetetraacetic acid (EDTA) as an anticoagulant and plasma for resuspension in platelet-rich plasma (PRP) preparation, demonstrating the benefits of this combination in terms of enhanced platelet recovery and preservation of platelet functionality.
Our study design was initially to investigate the effects of two different anticoagulants, namely sodium citrate (SC) and EDTA.. Following the determination that EDTA exhibited superior suitability as an anticoagulant, we proceeded to compare the resuspension media options of plasma versus sodium chloride (NaCl). Through this systematic approach, we concluded that the combination of EDTA as the anticoagulant and plasma as the resuspension medium outperforms other combinations, resulting in increased platelet recovery and preservation of platelet functionality during PRP preparation.
We have carefully considered your input and have detailed the methodology description to offer additional clarity on our study's approach.
2) In response to your recommendation, we have merged Figures 2 and 3 as per your suggestion.
3) In response to the concern about the sufficiency of experimental data to support our conclusions, we wholeheartedly agree with the importance of providing robust information to support our findings. Our study, in line with established methodologies in the literature, calculated platelet recovery rates as a percentage. We opted for this approach recognizing that incorporating absolute platelet concentration values, expressed as a mean, may not significantly enhance the comprehensiveness of our analysis.
The platelet recovery rate (PRR), calculated as
PRR (%) = (Platelets concentration in PRP × volume of PRP)/(Platelets concentration in whole blood × volume blood) × 100
refers to the relative increase in platelet concentration in PRP compared to the initial platelet concentration in whole blood. This methodological choice is particularly advantageous in studies involving biological samples with inherent variability among individuals, as it mitigates such variability by reporting the analysis to the initial count of platelets.
However, we appreciate and value your suggestion. We have conducted a slight revision of the subsection focusing on platelet concentration, aiming to ensure that the presentation of results provides clarity regarding the comparison groups involving anticoagulants and resuspension media. This adjustment has been made to enhance the readability and understanding of the study's findings, particularly concerning the influence of different variables on platelet concentration in PRP preparation.
Data regarding platelet yield is provided in section results - 3.1 and 3.2, including tables. However, for transparency and completeness, we may provide additional data. If you consider it necessary, we are open to incorporating this information into the final version of the paper.
Your feedback is invaluable, and we are committed to ensuring the integrity and clarity of our research.